# Challenges Facing Viral Hepatitis C Elimination in Lebanon

**DOI:** 10.3390/pathogens12030432

**Published:** 2023-03-09

**Authors:** Nour Ayoub, Taha Hatab, Abdul Rahman Bizri

**Affiliations:** 1Faculty of Medicine, American University of Beirut, Beirut 11-0236, Lebanon; 2 Faculty of Medicine, Division of Infectious Diseases, Department of Internal Medicine, American University of Beirut, Beirut 11-0236, Lebanon

**Keywords:** hepatitis C, infection, Lebanon, incidence, prevalence, elimination, challenges

## Abstract

Hepatitis C is a hepatotropic virus that causes progressive liver inflammation, eventually leading to cirrhosis and hepatocellular carcinoma if left untreated. All infected patients can achieve a cure if treated early. Unfortunately, many patients remain asymptomatic and tend to present late with hepatic complications. Given the economic and health burdens of chronic hepatitis C infection, the World Health Organization (WHO) has proposed a strategy to eliminate hepatitis C by 2030. This article describes the epidemiology of hepatitis C in Lebanon and highlights the challenges hindering its elimination. An extensive search was conducted using PubMed, Medline, Cochrane, and the Lebanese Ministry of Public Health–Epidemiologic Surveillance Unit website. Obtained data were analyzed and discussed in light of the current WHO recommendations. It was found that Lebanon has a low prevalence of hepatitis C. Incidence is higher among males and Mount Lebanon residents. A wide variety of hepatitis C genotypes exists among various risk groups, with genotype 1 being the most predominant. In Lebanon, many barriers prevent successful hepatitis C elimination, including the absence of a comprehensive screening policy, stigma, neglect among high-risk groups, economic collapse, and a lack of proper care and surveillance among the refugees. Appropriate screening schemes and early linkage to care among the general and high-risk populations are essential for successful hepatitis C elimination in Lebanon.

## 1. Introduction and Background

Hepatitis C virus (HCV) is a well-established cause of chronic hepatitis and end-stage liver disease [1]. Globally, an estimated 58 million individuals were living with chronic HCV infection in 2017, with an annual incidence of 1.5 million new cases. In 2019, more than 290,000 people died from HCV-related complications, mainly liver cirrhosis and hepatocellular carcinoma [2].

On a molecular level, HCV is an enveloped positive-sense RNA virus related to the *Flaviviridae* family [3]. There are seven known genotypes and at least eighty subtypes of the virus due to its error-prone replication process [4]. This extensive genetic variability of the virus represents a major barrier to a successful HCV vaccine and virus clearance.

HCV is a blood-borne virus primarily transmitted through contaminated blood and blood products [5]. Most individuals become infected through sharing contaminated needles and syringes during recreational drug injections. People can become infected in healthcare facilities due to the inappropriate sterilization of medical equipment, inadequate screening before transfusion, and improper disposal of sharps [6]. HCV can also spread via tattoos and piercings if non-sterile equipment is used [7]. Perinatal transmission from mothers to their infants may occur and is the leading cause of HCV infection in children. The reported rate of occurrence is approximately 6% in neonates born to HCV-infected mothers and approximately twice that in those born to HIV/HCV co-infected mothers [8]. Sexual transmission remains an uncommon route, but an increasing number of HCV cases are reported in HIV-infected men who have sex with men [9]. About 2.5 million (6.2%) HIV-infected individuals have an HCV infection. The chronic hepatic disease represents a major cause of morbidity and mortality among people living with HIV globally [1].

HCV can cause both acute and chronic infections. Approximately 25% of acutely infected individuals clear the virus spontaneously within 6 months without any treatment [10]. Those who do not clear the virus will develop chronic infection; eventually, 15–20% of those will have liver fibrosis and may progress to cirrhosis [11]. Once cirrhotic, the patient becomes at risk of acute liver decompensation at rates of 3–6% per year, which can present with life-threatening conditions, such as variceal hemorrhages, ascites, and hepatic encephalopathy. In end-stage liver disease, the risk of death in the following year is usually 15–20%. Patients with cirrhosis can also develop hepatocellular carcinoma at rates of 1–5% per year [12]. Several extrahepatic manifestations have been associated with chronic HCV, including mixed cryoglobulinemia vasculitis, B-cell lymphoproliferative diseases, Sicca syndrome, and cardiovascular diseases, among others [13].

The annual global economic burden of HCV exceeds $10 billion. This includes the cost of screening, antivirals, management of hepatic and extrahepatic complications, and loss of work productivity due to impaired quality of life [14,15]. This cost can be reduced by early detection and proper treatment. Essentially, all infected patients are expected to clear the virus if treated early. Current FDA-approved HCV treatment includes RNA-dependent RNA polymerase inhibitors, including NS3/4A protease, NS5A protein of HCV, ribavirin, and pegylated interferon. Typically, two specific inhibitors are given in combination; the usual duration of treatment is 12 weeks [16].

Despite the advances in medical therapy, HCV prevalence remains high in developing countries, with new cases on the rise [17]. The lack of a comprehensive screening plan and the scarcity of antiviral therapies are major barriers to successful elimination in these countries. Lebanon is a developing country along the eastern coast of the Mediterranean Sea, sharing its borders with Syria [18]. HCV prevalence in Lebanon (0.2%) is relatively comparable to that of neighboring countries (0.2% in Iraq, 0.3% in Jordan, 0.2% in Palestine, and 0.4% in Syria) and less than that of Egypt (14.7%) [19]. Among high-risk populations, Lebanon has a relatively lower incidence (14.5%) of HCV when compared to Iraq (19.5%), Jordan (37%), and Syria (47.4%) [19]. The conflicts in the Middle East and many neighboring countries have forced many groups to flee to Lebanon, which has increased the hepatitis C burden in the country [19]. Currently, 6 million individuals, including refugees, live in Lebanon [20]. Lebanon has the highest proportion (11%) of older adults aged 65 years and above among all Arab countries [21]. The United Nations estimated that the average life expectancy in Lebanon will increase to 78.7 by 2050, leading to an increase in the percentage of the older population to 25.8% [22]. Since most HCV-infected people in Lebanon are 40 to 60 years of age, the aging of the current HCV-infected population will contribute to a substantial increase in the economic burden in the coming years due to age-associated morbidity and mortality [23,24]. In a multi-cohort health-state-transition model published in 2018, it was projected that without adequate therapy, the number of patients diagnosed with compensated/decompensated cirrhosis, hepatocellular carcinoma, or liver-related death will increase significantly in 2036, especially among the elderly population [25]. This will have a huge economic impact on the country, estimated to be approximately 150 million EUR [25].

The WHO has set a target to eliminate viral hepatitis by 2030. The goals include reducing HCV incidence by 80%, diagnosing 90% of affected individuals, and treating 90% of those diagnosed by scaling up screening and treatment services [26]. In the absence of an effective vaccine, hepatitis C prevention depends on reducing exposure in high-risk populations and healthcare settings. Primary prevention interventions suggested by the WHO include educating healthcare workers about the safe handling and disposal of sharps, implementing harm-reduction strategies in people who inject drugs, and screening blood products for blood-borne infections.

## 2. Methodology

A cross-sectional descriptive analysis was conducted on the epidemiology, genotype distribution, and risk factors associated with hepatitis C infection in Lebanon. Challenges to HCV elimination were identified, and several recommendations were discussed to meet the WHO elimination goals by 2030. A comprehensive search of peer-reviewed articles was completed using the following MeSH and keywords: hepatitis C, Lebanon, refugee, incidence, and prevalence. Three databases were searched: PubMed (https://pubmed-ncbinlm-nih-gov; accessed on 1 February 2022), Medline (https://ovidsp-dc2-ovid.com; accessed on 1 February 2022), and Cochrane Library (www.cochranelibrary.com; accessed on 1 February 2022). Forty-three relevant studies were identified between the years 2005 and 2019, including systematic reviews, meta-analyses, abstracts, and case reports. In addition, data from the Lebanese Ministry of Public Health–Epidemiologic Surveillance Unit (LMoPH-ESU) website were analyzed and tabulated using Microsoft Excel.

## 3. HCV in Lebanon

### 3.1. Prevalence and Incidence

Several studies have evaluated the distribution of HCV in Lebanon across the years. In a study published in 2016, the seroprevalence was 0.21% [22]. HCV prevalence in Lebanon varies across governorates as well, with Nabatieh being the highest (0.61%), followed by Beirut (0.26%) [22]. As for HCV incidence, a total of 1333 new cases were reported between the years 2005 and 2019, with a male predominance of 74.5% [26]. Table 1 and Figure 1, Figure 2 and Figure 3 present the distribution of new cases by age, gender, and residence in Lebanon [26]. Most new cases belonged to the age group of 20–39 years (34.2%), were predominantly males (74.8%), and were mainly located in Mount Lebanon municipality (27.7%).

### 3.2. Routes of Transmission and High-Risk Groups

Intravenous drug use and blood transfusion are the most common routes of transmission, followed by hemodialysis [23,27]. Vertical and sexual transmissions are the least common routes of infection, representing 0.2% and 0.3% of HCV transmission cases, respectively (Table 2) [28]. Several studies have reported HCV prevalence among people who inject drugs (PWID) and prisoners. In a recent study conducted in 2020, HCV prevalence among 250 PWID was 15.6% [28], while previous studies showed a higher rate, reaching 52.8% [19]. In another small study, HCV prevalence among prisoners was 3.4% [29], with a higher rate of 28.1% reported a few years later in a larger study [30].

### 3.3. Genotype Distribution of HCV among High-Risk Groups

At the national level, the most common HVC genotype is 1, followed by genotype 4 [19,27,31,32,33]. Genotype 3 is the most prevalent among PWIU and prisoners [19,27,31,32,33]. In thalassemic patients, genotype 4 has been found to account for 35–50%, followed by genotype 1 (20–30%) [34,35,36]. Several studies have described genotype distribution among hemodialysis patients in Lebanon [19,27,36,37]. In a small study, genotype 2 accounted for 80% [36]. In a more recent report on 183 HCV patients, genotype 1 represented 62% [27].

### 3.4. Therapy Prescription

The MoPH has a drug assistance program that offers free DAA to all Lebanese people across the nation [25]. To qualify for the program, the patient has to be referred to a specialist (an infectious disease or gastroenterologist) who examines the patient and fills out the necessary forms documenting the mode of transmission of the disease, stage of illness, genotype, coexisting viral infections, and previous therapies. The medications are then dispensed by special centers affiliated with the MoPH. After starting treatment, the patient is followed up periodically to assess the virological response and disease progression.

### 3.5. Barriers to HCV Elimination

#### 3.5.1. Absence of a National Screening Plan

The majority (75%) of those infected with HCV remain asymptomatic for years after exposure and tend to seek medical care only when liver-related complications arise [38]. In a study conducted in Lebanon between 2014 and 2016, almost 65.5% of the newly diagnosed HCV infections were in adults older than 40 years of age who presented with moderate to high levels of liver fibrosis [24]. According to the health-state transition model mentioned previously, introducing DAAs at such advanced stages of fibrosis will lead to fewer life years spent in sustained virologic response when compared to treatment at earlier stages [25]. It was also shown that early introduction of DAAs in the course of the disease improved survival and reduced liver-related and extrahepatic complications. Thus, the absence of early screening protocols represents a major barrier to the early detection and elimination of HCV, despite the availability of highly effective oral anti-HCV agents in the country.

#### 3.5.2. Intravenous Drug Use and Associated Stigma

Intravenous drug use remains a serious public health problem in Lebanon, as it increases the risk of blood-borne infections. Compared to the general population, people who inject drugs (PWID) are the most common subpopulation infected with hepatitis C [28,33]. Of the HCV-positive drug users, 53.8% have been found to have a history of sharing unsterile syringes because of the knowledge barrier and/or limited access to sterile equipment. Almost 21.6% of HCV-positive drug users have been found to be unaware of the possibility of transmission when using unsterile needles [28]. On the other hand, even those who are aware of the complications face difficulties obtaining sterile equipment. Cultural stigma is a substantial barrier to safe drug-use practices in Lebanon. Society often labels drug users as criminals, which limits their access to sterile syringes. Such a stigma exists among healthcare professionals as well. In one study carried out in 2017, pharmacists refused to dispense sterile needles and alcohol pads to avoid contributing to further harm and addiction [39]. This would encourage PWID to reuse contaminated syringes. So far, NGOs are running all needle-exchange programs and the bulk of awareness campaigns with minimal governmental support and, thus, have very limited financial and human resources. They are unable to meet the needs of all PWID across the country [39].

#### 3.5.3. High-Risk Prison Environment

Incarceration is another major risk factor for HCV transmission; the risk of infection is directly proportional to the length of stay in prison [30]. Prisoners in Lebanon have higher prevalence rates of HCV compared to the general population [29]. Histories of previous incarceration and drug injection were found to be independent risk factors for HCV infections among prisoners. Many prisoners had a history of intravenous drug use before their incarceration, and many continued to inject drugs in prison under non-sterile conditions [30]. Tattooing and the use of non-sterile injection equipment in prison were also associated with HCV infections [29,30]. In addition, poor prison conditions facilitate HCV transmission among prisoners. It is estimated that Roumieh prison, which is one of the largest in Lebanon, accommodates at least three to four times the capacity it was originally designed for [40]. Hygiene standards in some Lebanese prisons are also suboptimal due to restricted access to showers and the absence of soap [40].

#### 3.5.4. Refugees

Lebanon is the country with the highest number of refugees per capita in the world [41]. The presence of many illegally residing Syrian and Palestinian refugees in the country adds to the complexity of the problem. Unfortunately, the Lebanese government provides free HCV treatment to Lebanese citizens only, leaving at least 30% of the population without access to therapy.

#### 3.5.5. Economic Collapse

Early anti-viral therapy initiation is essential to eliminate HCV and prevent chronic complications. Until 2019, Lebanon ranked among the top Middle Eastern health systems on the Healthcare Access and Quality (HAQ) index [42]. The LMoPH provided free antiviral medications to HCV patients [25]. With the recent political turmoil and the economic crisis, that is no longer the case. Due to catastrophic economic conditions, all healthcare sectors, including primary care centers, pharmaceuticals, and medical supply industries, were crippled by the economic downturn [43]. With the devaluation of the local currency, which lost more than 12 times its value, the LMoPH lost most of its financial assets. The withdrawal of government subsidies on many medications rendered the vast majority of the Lebanese population unable to obtain or purchase essential medications, mainly for chronic illnesses, including viral hepatitis.

## 4. Discussion

The three pillars of HCV elimination include screening, diagnosing, and linking to care. In the absence of an effective HCV vaccine, the best approach to eliminate the disease is through prevention and treatment. For more than two decades, the main treatment for HCV was a combination of poorly tolerated IFN-y and ribavirin, with a cure rate ranging between 40 and 65% [44,45]. However, with the current availability of effective oral antiviral agents with a sustained virologic response and viral cure rates of 92–100%, HCV elimination is an achievable goal [46].

Lebanon has a relatively low incidence of HCV infection (Table 1, Figure 1), with a high variation in genotypes, which ranks Lebanon high in the Arab world on the Shannon Diversity Index, a score that assesses diversity in the types and frequency of isolated HCV genotypes [47,48]. A low score suggests that the majority of infections are associated with one or two dominant genotypes, whereas a high score shows that infections are more evenly spread among several genotypes [47]. Such a finding can be explained by the large-scale emigration in Lebanon, as population movements provide avenues for genotypes prevalent in other parts of the world to be introduced into the local circulation. The higher prevalence of HCV infection in some municipalities in Lebanon can be justified by socioeconomic conditions, contamination-risky religious rituals, frequent immigration to HCV-endemic areas, as well as sequelae of repeated wars in Southern Lebanon [23].

To prevent HCV infection and meet the WHO elimination targets, policies should aim at the main modes of transmission, focusing on high-risk groups, mainly PWID, those incarcerated, hemodialysis patients, individuals who received multiple blood transfusions before 1992, people living with HIV (PLHIV), those who engage in unprotected sexual activities, and others [49]. An enhanced screening policy coupled with facilitated access to direct antivirals will diminish the future burden of HCV and provide health benefits with net cost savings [25].

A recent multi-cohort health-state-transition model showed that chronic HCV will have a huge economic burden on Lebanon by 2036 in the absence of adequate screening and treatment strategies [25]. Because testing modalities are cheap, screening strategies should be adopted at the population level to detect asymptomatic individuals and allow proper interventions before complications occur. The Centers for Disease Control and Prevention (CDC) recommends screening all adults 18 years and over at least once during their lifetime, while high-risk individuals—such as those who currently inject drugs, have ever injected drugs, have HIV, or are on hemodialysis—should be screened regularly [49]. The CDC also recommends screening those who received blood products or organs before 1992 and who received clotting factor concentrates before 1987 [49]. In Lebanon, current screening strategies to prevent HCV include blood and blood product screening, biannual screening for hemodialysis patients, pre-marital testing, and foreign laborer screening [50,51]. Lebanon adopted HCV screening for all blood and blood product donors in 1992, which helped lower the incidence rate of the infection in the country; however, no guidelines exist to screen those who received transfusions before that time, mainly during the Lebanese civil war, which started in 1975 [51,52]. Screening individuals aged 30 years and over (2,193,639 individuals) and those who received blood transfusions due to the war between 1975 and 1992 would be a cost-effective approach to detecting asymptomatic individuals (Table 1, Figure 1 and Figure 4). However, recent reports by the LMoPH showed that most new HCV cases between 2005 and 2019 are in the age group of 20 to 35 years, which makes screening those aged 18 years and over (3,195,132 individuals) a more appropriate approach (Table 1, Figure 1 and Figure 4).

Globally, almost 52.3% of PWID are anti-HCV positive [53]. Frequent testing and proper education about the modes of transmission are adopted by many countries to minimize HCV contraction. PWID who test positive are then linked to medical care. It was found that even individuals who are currently IV drug users can still benefit from HCV treatment, with the treatment response being similar to that of the rest of the population [54]. A main barrier to care in this high-risk group is that members of this group are often hard to reach. The association of HCV with certain high-risk behaviors leads to stigmatization by blaming those affected for contracting the disease [55]. The link between HIV and HCV is another reason for potential isolation and stigmatization. Stigma may hurt one’s self-esteem and quality of life and can make access to medical care more difficult, increasing the chance of chronic infection remaining undiagnosed [55]. Healthcare workers are not immune to preconceptions and judgments, which affects their performance when dealing with HCV patients [56]. Addressing this issue can help patients avoid isolation, increase compliance, and encourage the seeking of medical assistance. Strategies to eliminate HCV in PWID in Lebanon should target stigma; implement harm reduction services, such as needle-exchange programs; and provide affordable medical care. HCV counseling and screening should be provided in settings accessible to PWID, such as opiate-substitution-therapy providers, hospitals, NGOs, prisons, and needle-exchange sites [53]. All individuals who test positive should receive treatment and be followed up routinely. Lebanon has succeeded in reducing the incidence of HIV in PWID by adopting such measures, which can be an indicator for lowering incidence rates when projecting to HCV.

Prison constitutes a high-risk environment due to unsafe drug injection, sex, and tattooing [57]. Currently, no screening protocols are implemented for HCV screening in Lebanese prisons. Regular HCV screening and treatment can be cost-effective in such an environment, as it reaches high-risk populations, such as PWID, who are usually over-represented in prisons [58]. Counseling programs to educate prisoners about different modes of transmission and prevention should be implemented as well (Table 3). Harm reduction strategies, such as providing drug users with sterile needles and cleaning equipment, and educating them about safe sexual measures, should be adopted to decrease further disease transmission. Enrolment in support groups and rehabilitation facilities to ease the life shift can also be employed [59]. Prisoners should also be supported once released, as many find it difficult to find a job, which leads to financial hardship, eventually causing the de-prioritization of medical treatment, including that for HCV [60].

ILO: International Labor Organization, NGO: Non-Governmental Organization, PPE: Personal Protective Equipment, PEP: Post-Exposure Prophylaxis, UNRWA: United Nations Relief and Works Agency, UNHCR: United Nations High Commissioner for Refugees.

Healthcare workers (HCWs) are at risk of contracting HCV through syringes, sharps, and other hazardous materials [63]. A study conducted at the American University of Beirut Medical Center found that, between 2014 and 2018, the mean blood-borne pathogen exposure incidence rate was 5.4 per 100 full-time employees, 65.6 per 100 bed-years, and 0.48 per admission years [66]. Medical staff should use the appropriate personal protective equipment and universal precautions when handling body fluids [63]. Training on safe injection practices, waste management, and sharps disposal should be conducted regularly in medical centers [63]. Post-exposure protocols for management should be adopted as per the guidelines [64,65].

The disease-reporting system is more reliable in the Lebanese population compared to that of refugees for several reasons [26]. Many of the refugees are afraid to seek medical care because of the possibility of being reported for illegal residence. Refugees, then, are usually not linked to care once they test positive [62]. Complete statistical data on HCV status among refugees is unavailable, which constitutes a major barrier to effective HCV elimination [26]. Given the current situation, the Lebanese government should apply the same screening and preventive strategies in the refugees as those implemented in the Lebanese population (Table 3). The government should dedicate more efforts to encouraging refugees to seek medical care, properly screen them, and link those who test positive to the appropriate care in coordination with international organizations, such as the United Nations High Commissioner for Refugees (UNHCR) and the United Nations Relief and Works Agency for Palestine Refugees (UNRWA).

HCV elimination requires a broad-based educational effort to improve knowledge of this condition and reduce negative stereotypes [67]. Patients, their families, healthcare providers, and society at large should be targeted. A nationwide awareness campaign should be conducted through widely used social media platforms to educate people about disease transmission, its silent nature, and the importance of screening.

The lack of substantial financial resources remains a major barrier to effective HCV elimination in Lebanon. More governmental support for the health sector is needed to achieve screening and treatment requirements and goals. Lebanon is experiencing an unprecedented economic crisis, which has forced the country to take strict expenditure-tightening measures. These measures should not compromise elimination strategies since early detection and treatment will not only increase life expectancy but will also decrease the economic burden on the country in the long term [68]. Given the current financial collapse, which affects all sectors in Lebanon, including healthcare, it would be practically difficult to allocate the needed governmental budget required for elimination [43]. National health authorities should seek the support of the international community and organizations to meet the WHO elimination goals.

## 5. Conclusions

Lebanon is a country with a low HCV prevalence. Several measures and actions are needed to achieve the WHO elimination targets by 2030. Major obstacles include the absence of a comprehensive screening policy and the lack of any national policy regarding refugee populations, incarcerated individuals, and the PWID community. Appropriate screening based on the local epidemiology of the infection, linkage to care, DAA initiation at early stages, and proper education are the mainstays of a successful elimination program. Austere economic conditions and financial collapse may hinder any immediate progress towards elimination and increase the HCV disease burden.

## Figures and Tables

**Figure 1 pathogens-12-00432-f001:**
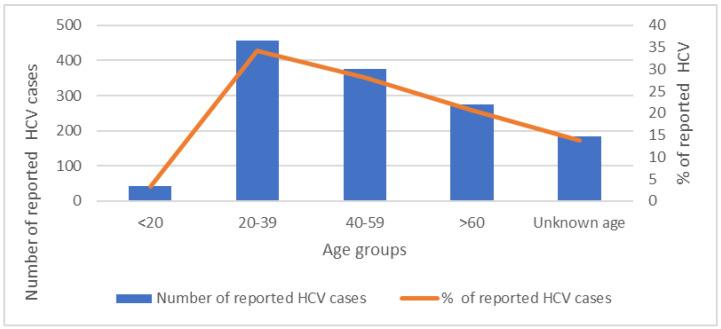
Age distribution of reported HCV cases between 2005 and 2019. Ministry of Public Health. (n.d.). Epidemiological Surveillance. Esu. https://www.moph.gov.lb/en/Pages/2/193/esu (accessed on 11 December 2021).

**Figure 2 pathogens-12-00432-f002:**
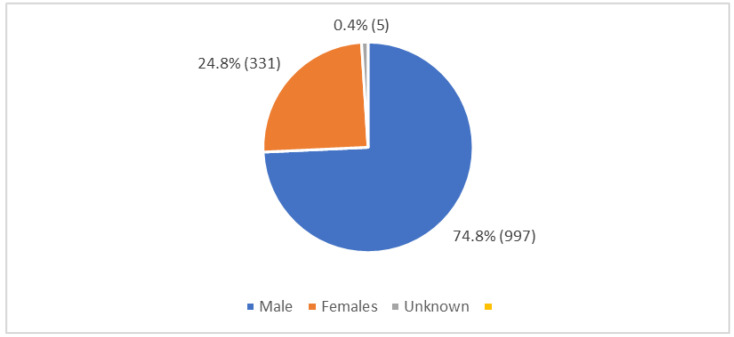
Distribution of reported HCV cases by gender between 2005 and 2019. Ministry of Public Health. (n.d.). Epidemiological Surveillance. Esu. https://www.moph.gov.lb/en/Pages/2/193/esu (accessed on 11 December 2021).

**Figure 3 pathogens-12-00432-f003:**
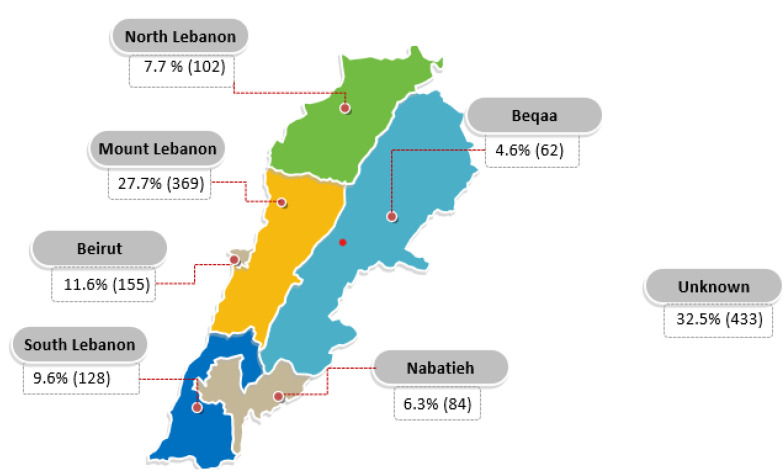
Distribution of reported HCV cases by province between 2005 and 2019. Ministry of Public Health. (n.d.). Epidemiological Surveillance. Esu. https://www.moph.gov.lb/en/Pages/2/193/esu (accessed on 11 December 2021).

**Figure 4 pathogens-12-00432-f004:**
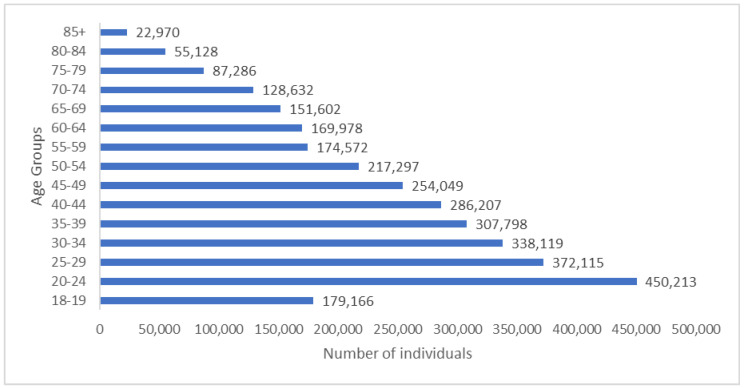
Estimated Lebanese population +18 years old in 2020. Central Administration of Statistics –. (n.d.). Population Statistics—2020. Demographic and Social Statistics. Accessed on 14 December 2021 from http://cas.gov.lb/index.php/demographic-and-social-en/population-en.

**Table 1 pathogens-12-00432-t001:** Age distribution of reported HCV cases between 2005 and 2019 *.

	<20 Years	20–39 Years	40–59 Years	>60 Years	Unknown Age	Total
HCV Number	43	456	376	275	183	1333
HCV %	3.2%	34.2%	28.2%	20.6%	13.7%	

* Ministry of Public Health. (n.d.). Epidemiological Surveillance. Esu. https://www.moph.gov.lb/en/Pages/2/193/esu (accessed on 11 December 2021).

**Table 2 pathogens-12-00432-t002:** Genotype distribution among different risk groups from Abou Rached et al. *.

Genotype	PWID	Hemodialysis	Transfusion	Sexual	Vertical	Unknown	Total
1	34.5%	61.7%	56.3%	0	100%	23.1%	47%
2	6.9%	2.2%	1.4%	0%	0%	2.1%	3.7%
3	37.9%	0%	3.5%	0%	0%	5.2%	14.1%
4	17.2%	36.1%	38.7%	100%	0%	24.7%	33.9%
5	3.4%	0.0%	0%	0%	0%	0.8%	1.4%
Total	28.1%	17.7%	27.5%	0.3%	0.2%	26.1%	1031

* Rached, A. A., Nakhoul, M., Richa, C., Jreij, A., Hanna, P. A., and Ammar, W. (2020). Prevalence of hepatitis B and anti-hepatitis C virus antibodies among people who inject drugs in the Lebanese population. Eastern Mediterranean health journal: La revue de sante de la Mediterranee orientale: al-Majallah al-sihhiyah li-sharq al-mutawassit, 26(4), 461–467. https://doi.org/10.26719/emhj.19.094; accessed on 11 December 2021 [28].

**Table 3 pathogens-12-00432-t003:** Current strategies and suggested recommendations to eliminate HCV in Lebanon.

Targeted Population	Current Strategies	Recommendations	References
Community at large	- Premarital screening. - No current universal screening.	- At least once, screen people ≥ 18 years old. - Educate about the nature of the disease, modes of transmission, and the availability of curative treatment.	[49,50,51,52] Table 1 Figure 1 and Figure 4
Incarcerated individuals	- No current HCV screening policy.	- Screen regularly. - Link those who test positive to care. - Improve prison conditions (hygiene). - Educate about modes of transmission and safe needle use and safer sexual practices. - Establish needle-exchange programs. - Provide post-release support. - Screen guards and employees working at prisons.	[49,57,58,59,60]
PWID	- NGO-based counseling centers. - NGO-based needle distribution.	- Implement strategies to reach PWIDs. - Screen regularly. - Link those who test positive to care. - Educate about modes of transmission and safe drug (needles) use, social behaviors, and sexual practices. - Establish needle-exchange programs. - Eliminate stigma by raising awareness among the general population and medical staff.	[49,53,54,55,56]
Foreign working force	- Screen prior to work permit.	- Link positive individuals to care in collaboration with ILO.	
Refugees	- No available data.	- Apply same screening and preventive strategies as those of the Lebanese population. - Work closely with UNRWA, UNHCR, and other NGOs to screen and treat positive individuals.	[61,62]
Hemodialysis	- Screen patients twice annually.	Keep same.	[49]
Blood bank	- Screen blood products and blood.	Keep same. Encourage self-deferral.	[49]
Healthcare workers	- Provide appropriate PPE. - Abide by universal precautions. - Train on safe injection practices and sharps disposal.- Provide PEP.	Keep same.	[49,63,64,65]

## Data Availability

Not applicable.

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
