# Peer review of "Challenges Facing Viral Hepatitis C Elimination in Lebanon"

_pathogens, 2023, doi:10.3390/pathogens12030432_

Round 1

Reviewer 1 Report (Previous Reviewer 2)

The topic of your manuscript is interesting and I think your work is worth publishing, but you have serious flaws in the organization of the article. You note that your article is a review, but you have organized it as a research paper with results.

Organize your manuscript as a review.

3.1. Prevalence and Incidence of HCV- Please note that these data are for Lebanon

L117 Reference 26 I believe is not correctly cited

L 118 I do not see table 1, nor figures!

I don't see table 2 either, which leads me to believe that these tables are in the cited article.

You should make your own tables and figures to present your research summary.

In general, you have serious gaps in the presentation of the topic you are researching.

Author Response

Response to Reviewer 1

Thank you for your comments. We have gone through your comments carefully and tried our best to address them one by one. We hope the manuscript has been improved accordingly.

  1. The topic of your manuscript is interesting and I think your work is worth publishing, but you have serious flaws in the organization of the article. You note that your article is a review, but you have organized it as a research paper with results. Organize your manuscript as a review.

We revised the different sections and reorganized the manuscript as a review.

  1. 3.1. Prevalence and Incidence of HCV- Please note that these data are for Lebanon

    Noted as advised.

  1. L117 Reference 26 I believe is not correctly cited.

Thank you for pointing this out. We adjusted the citation based on the WHO’s website citation as follows: “World Health Organization. (‎2016)‎. Global health sector strategy on viral hepatitis 2016-2021. Towards ending viral hepatitis. World Health Organization. https://apps.who.int/iris/handle/10665/246177”

  1. L 118 I do not see table 1, nor figures! I don't see table 2 either, which leads me to believe that these tables are in the cited article. You should make your own tables and figures to present your research summary.

Thank you so much for pointing this out. Our graphs & tables were not uploaded with the manuscript. We will attach them with the current submission.

  1. In general, you have serious gaps in the presentation of the topic you are researching.

Thank you for your comment. We hope that we have addressed those gaps in the current revision. If not, we would appreciate any further suggestions. 

Note: we went through the entire manuscript and made minor grammatical adjustments.

Reviewer 2 Report (New Reviewer)

Ayoub et al. submitted their review article discussing the challenges of viral hepatitis C elimination in Lebanon. Generally speaking, the paper is quite well-written. I raise some minor concerns,

1.     Line 64-66: please add pegylated interferon which is also the approved medication for CHC.

2.     Line 144-145: please describe more about the treatment of HCV by DAAs and the efficacy in the country

3.     In the conclusion part, The DAAs therapy has to be mentioned.

4.     The condition of the HCV RNA tests, the locations of the treatment,  and the qualification of the treaters, etc have to be described in Lebanon.

Author Response

Response to Reviewer 2

Thank you for your comments. We have gone through your comments carefully and tried our best to address them one by one. We hope the manuscript has been improved accordingly.

  1.  Line 64-66: please add pegylated interferon which is also the approved medication for CHC.

Thank you for pointing this out. We revised the sentence (lines 64-66) accordingly.

  1. Line 144-145: please describe more about the treatment of HCV by DAAs and the efficacy in the country.

Thank you for your comment. We revised the referred sections to include the following:

“According to the multi-cohort health-state transition model mentioned previously, introducing DAAs at such advanced stages of fibrosis will lead to fewer life-years spent in sustained virologic response. It was also shown that early introduction of DAAs in the course of the disease improved survival, and reduced liver-related and extrahepatic complications.”

  1. In the conclusion part, The DAAs therapy has to be mentioned.

We revised the conclusion to include DAA.

  1. The condition of the HCV RNA tests, the locations of the treatment,  and the qualification of the treaters, etc have to be described in Lebanon.

Thank you for pointing this out. Unfortunately, limited data is available on the HCV RNA tests status in Lebanon. However, we added a section under “HCV in Lebanon” as “Therapy Prescription” that includes the following:

“The MoPH has a drug assistance program that offers free DAA to all Lebanese across the nation [25]. To qualify for the program, a specialist has to examine the patient has to be referred to a specialist (an infectious disease or gastroenterologist) who examines the patient and fills out the necessary forms documenting the mode of transmission of the disease, stage of illness, genotype, coexisting viral infections, and previous therapies. The medications are then dispensed by special centers affiliated with the MoPH. After starting treatment, the patient is followed up periodically to assess the virological response and disease progression.”

Note: we went through the entire manuscript and made minor grammatical adjustments.

Round 2

Reviewer 1 Report (Previous Reviewer 2)

Tablicite and figures are again not inserted in the text and I don't see them as additional files.

Please include your tables and figures.

Author Response

Thank you for reviewing our manuscript.

We reached out to the editor, and we think you are able to see the tables and figures now.

Thank you for your patience.

Best,

Taha Hatab, MD

Round 3

Reviewer 1 Report (Previous Reviewer 2)

In the introduction, I would like to see data on the global prevalence of HCV and also the prevalence by country. You could present a graph of the percentage prevalence of HCV in different countries like Pakistan, India, Egypt, China, Russia, etc. I would like you to evaluate the epidemiological data over the years, and how they are improving.

I would advise you to rearrange the information in the introduction and start strong. Your first sentence in the introduction is not the most appropriate for the specific problem you are addressing.

In the results, you told: L116 The most recent study was in 2016 when the estimated seroprevalence was 0.21% . After that L145: In a recent study conducted in 2020, HCV prevalence 146 among 250 PWID was 15.6% [28], while previous studies showed a higher rate, reaching 147 52.8% [19].

Please, clarify the information in L116, delete "most recent", or include total seroprevalence.

Please explain the results of Figure 3. Explain how the percentages were calculated because these high percentages are confusing.

Will be good if you include the list of the used abbreviations. 

Author Response

Response to Reviewer 1

Thank you for your comments. We have gone through your comments carefully and tried our best to address them one by one. We hope the manuscript has been improved accordingly.

Point 1: In the introduction, I would like to see data on the global prevalence of HCV and also the prevalence by country. You could present a graph of the percentage prevalence of HCV in different countries like Pakistan, India, Egypt, China, Russia, etc. I would like you to evaluate the epidemiological data over the years, and how they are improving.

Answer: Thank you so much for your input. Although such information is crucial in evaluating HCV burden globally, we think that presenting such data would deviate the compass from our main theme which is Lebanon and the challenges facing Lebanon in elimination of HCV.

However, we did present HCV prevalence from neighboring Middle Eastern countries (L73-L81) because as we explained in the manuscript, these countries had a direct/indirect effect on disease burden in Lebanon

Point 2: I would advise you to rearrange the information in the introduction and start strong. Your first sentence in the introduction is not the most appropriate for the specific problem you are addressing.

Answer: Thank you for your comment. We edited our introduction accordingly, and we started by HCV burden globally before dwelling into the molecular level of the virus.

"Hepatitis C virus (HCV) is a well-established cause of chronic hepatitis and end-stage liver disease [1]. Globally, an estimated 58 million individuals were living with chronic HCV infection in 2017, with an annual incidence of 1.5 million new cases. In 2019, more than 290,000 people died from HCV-related complications, mainly liver cirrhosis and hepatocellular carcinoma [2]. On a molecular level, HCV is an enveloped positive-sense RNA virus related to the Flaviviridae family [3]. 

Point 3: In the results, you told: L116 The most recent study was in 2016 when the estimated seroprevalence was 0.21% . After that L145: In a recent study conducted in 2020, HCV prevalence 146 among 250 PWID was 15.6% [28], while previous studies showed a higher rate, reaching 147 52.8% [19]. Please, clarify the information in L116, delete "most recent", or include total seroprevalence.

Answer: Thank you for your comment. We removed "the most recent"

The study published in 2016 examined the overall seroprevalence in Lebanon (0.21%) while the study published in 2020 examined the seroprevalence among PWIDs which was had conflicting data between 2 studies (1st was 15.6% while the second was 52.8%)

Point 4: Please explain the results of Figure 3. Explain how the percentages were calculated because these high percentages are confusing.

Answer: The data in the Figures 1-3 are from the Lebanese Ministry of Public Health website. It was cited in text under each figure, and it was referenced accordingly. Figure 3 represents the % of new cases (L122) by area of residence/municipality. We added one sentence to briefly describe Figures 1, 2 and 3:

Most new cases belonged to the 20-39 years age group (34.2%), were predominantly males (74.8%), and were mainly located in Mount Lebanon municipality (27.7%).

Relevant abbreviations were added after Keywords in the first page. All abbreviations were cited in text

Note: we went through the entire manuscript and made minor grammatical adjustments.

This manuscript is a resubmission of an earlier submission. The following is a list of the peer review reports and author responses from that submission.

Round 1

Reviewer 1 Report

The authors review the status of hepatitis C virus infection in Lebanon, and the challenges facing the implementation of a successful national elimination program by 2030. Apart from the medical data, the review immerses the reader into cultural and historic facts. However, there are some issues that need to be resolved:

  • The introduction is loose. The authors present information that is already known like the type of the virus, its transmission mode or the types of infection that can cause. The information is beyond the scope of the article. Authors should detail WHO policies on reducing HCV by 2030 and the impact of HCV in Middle East and Lebanon.
  • The English needs polishing

Author Response

Response to Reviewer 1:

  • We re-wrote the introduction, we cut some of the info related to the general microbiology and pathophysiology of HCV infection. We added more information about Lebanon and neighboring countries (mainly Syria) to better fall within the scope of the article. We also addressed people co-infected with HIV. We tackled the issue of pharmacological therapy for HCV. Finally, we addressed WHO 2030 recommendations regarding HCV elimination and global burden of the disease.

Reviewer 2 Report

Authors: Nour Ayoub, Taha Hatab and Abdul Rahman Bizri

Title: Challenges Facing Viral Hepatitis C Elimination in Lebanon

The topic of the manuscript is relevant and would be of interest to the scientific community.

The article has too many technical and grammatical errors to be published. My recommendation to the authors to completely edit their work and resubmit it again.

  1. The abstract is not well structured and needs to be rewritten. There are a lot of grammatical errors throughout the text. It is not necessary to write the individual sections of the abstract (aim, methodology, results), etc.
  2. Line 22-23 There is no national elimination strategy in the country despite the free availability of antiviral treatment for the Lebanese. Several high-risk groups are not targeted, and the Syrian refugee population is not monitored or linked to care.

 For me, this information is linked more to discussion than a result.

  1. Line 24-25 Lebanon is a country with low viral hepatitis C prevalence. Lebanon ranks high in the Arab world on the Shannon Diversity Index, a score that assesses diversity in the types and frequency of isolated HCV genotypes.

The Shannon Diversity Index needs to be explained.

My recommendation is to rewrite the whole abstract, presenting the information much more focused.

  1. The introduction does not provide sufficient information on the subject. I do not see data on the spread of hepatitis C in the countries bordering Lebanon and about the confirmed HCV drug therapy; include the information about co-infection with HIV
  2. Write italic for Flavaviridae family
  3. You need to include additional information in Materials and Methods such as a Shannon Diversity Index; give the links to the PubMed, Medline, and Cochrane Library; determine the period for which the study was performed (data published between…. year to year).

Results

  1. In order to better organize the results, I suggest you put subtitles in the results section.
  2. Please include the information about how many studies are included in the analysis.
  3. In order to better present the results of the distribution of HCV in the different regions of Lebanon; please make a figure with the map of Lebanon, the regions and the distribution of HCV by regions with data from table 3.
  4. The figures and tables should be numbered consecutively in the text, and should be inserted as close as possible to their citation. I see first Table 2 and Figure 2 and after that Figure 1. Please change the numbering!
  5. Table 4: Dou you have IPR permission to use the data?
  6. Line 136 Several studies described genotype distribution among hemodialysis patient in Lebanon. Include the references!
  7. Line 187-193 The information is redundant and may be shortened
  8. Line 194-218 The paragraph should be rewritten for a clearer presentation of the results of the research and their differentiation from the discussion.

Discussion

  1. In general, I like the information presented in the discussion. However, it is necessary to correct the numbering of tables and figures and use the help of a colleague with a native English language  and make the text easier to understand.

Author Response

Response to Reviewer 2:

  • The whole abstract was re-written
  • Line 22-23 “There is no national elimination strategy in the country despite the free availability of antiviral treatment for the Lebanese. Several high-risk groups are not targeted, and the Syrian refugee population is not monitored or linked to care” was moved to the discussion.
  • Shannon Diversity index was explained in the discussion. Please note that we didn’t calculate the score because it requires a specific complicated formula along with some variables that are not provided in this review paper. We took this information from Messina et al. who already did the extensive mathematical calculations (with references of course).
  • Introduction was more or less re-written, points like “hepatitis C in the countries bordering Lebanon and about the confirmed HCV drug therapy; include the information about co-infection with HIV” were added and addressed.
  • Flavaviridae was written in Italic
  • Methodology was fixed, we added the corresponding websites for our search engines along with the number of studies and the period of the studies. Shannon Diversity Index formula wasn’t added because we didn’t calculate the index (explained in point 3)

Results:

  • Subtitles were added.
  • Number of studies included in the analysis were mentioned in the discussion.
  • Figure map of Lebanon was added, along with the prevalence of HCV in each province.
  • Figures and tables are consecutively added.
  • We took permission for including the table (from Rached et al.)
  • Info about mentioning references, it is fixed now.
  • Redundant information was removed.
  • “Line 194-218 The paragraph should be rewritten for a clearer presentation of the results of the research and their differentiation from the discussion” It was rewritten

Round 2

Reviewer 2 Report

The authors have significantly improved the quality of the manuscript, but I still see many errors and inaccuracies (p219, table 1. is repeated), but I am more worried about the interpretation of the results.

Page 67 HCV prevalence in Lebanon is relatively lower (0.21%) than that of neighboring countries (up to 9%).

First of all, this sentence does not say whether it is in the general population or in a certain group of people. The authors mentioned reference 19. The Epidemiology of Hepatitis C Virus in the Fertile Crescent: Systematic Review and Meta-Analysis, https://doi.org/10.1371/journal.pone.013528. Thuis article shows various analyzes, commenting on HCV prevalence among the general population and  populations at high risk. Their meta-analyzes estimated HCV prevalence among the general population at 0.2% in Iraq (range: 0–7.2%; 95% CI: 0.1–0.3%), 0.3% in Jordan (range: 0– 2.0%; 95% CI: 0.1–0.5%), 0.2% in Lebanon (range: 0–3.4%; 95% CI: 0.1–0.3%), 0.2% in Palestine (range: 0–9.0%; 95% CI) : 0.2–0.3%), and 0.4% in Syria (range: 0.3–0.9%; 95% CI: 0.4–0.5%). Among populations at high risk, HCV prevalence was estimated at 19.5% in Iraq (range: 0–67.3%; 95% CI: 14.9–24.5%), 37.0% in Jordan (range: 21–59.5%; 95% CI: 29.3 –45.0%), 14.5% in Lebanon (range: 0–52.8%; 95% CI: 5.6–26.5%), and 47.4% in Syria (range: 21.0–75.0%; 95% CI: 32.5–62.5%). Genotypes 4 and 1 appear to be the dominant circulating strains. This study shows that Lebanon's border countries have similar HCV prevalence among the general population

  1. 2% in Iraq
  2. 3% in Jordan
  3. 2% in Palestine
  4. 4% in Syria

Please, correctly describe the prevalence of HCV prevalence among the general population in Lebanon and  Lebanon's neighboring countries! Everywhere in the text, try to clear up the wrong information and related comments.

Please, include the date of HCV prevalence among populations at high risk in Lebanon and compared it with data from neighboring countries.

My decision is to Reconsider after major revision!

Author Response

Reviewer 2:

  • The whole abstract was re-written
  • Line 22-23 “There is no national elimination strategy in the country despite the free availability of antiviral treatment for the Lebanese. Several high-risk groups are not targeted, and the Syrian refugee population is not monitored or linked to care” was moved to the discussion.
  • Shannon Diversity index was explained in the discussion. Please note that we didn’t calculate the score because it requires a specific complicated formula along with some variables that are not provided in this review paper. We took this information from Messina et al. who already did the extensive mathematical calculations (with references of course).
  • Introduction was more or less re-written, points like “hepatitis C in the countries bordering Lebanon and about the confirmed HCV drug therapy; include the information about co-infection with HIV” were added and addressed.
  • Flavaviridae was written in Italic
  • Methodology was fixed, we added the corresponding websites for our search engines along with the number of studies and the period of the studies. Shannon Diversity Index formula wasn’t added because we didn’t calculate the index (explained in point 3)

Results:

  • Subtitles were added.
  • Number of studies included in the analysis were mentioned in the methodology.
  • Figure map of Lebanon was added, along with the prevalence of HCV in each province.
  • Figures and tables are consecutively added.
  • We took permission for including the table (from Rached et al.)
  • Info about mentioning references, it is fixed now.
  • Redundant information was removed.
  • “Line 194-218 The paragraph should be rewritten for a clearer presentation of the results of the research and their differentiation from the discussion” It was rewritten

Discussion:

  • We revised the discussion section and made some structural and linguistic changes.

Round 3

Reviewer 2 Report

The authors have significantly improved the quality of the manuscript, but I still see many errors and inaccuracies (p219, table 1. is repeated), but I am more worried about the interpretation of the results.

Page 67 HCV prevalence in Lebanon is relatively lower (0.21%) than that of neighboring countries (up to 9%).

First of all, this sentence does not say whether it is in the general population or in a certain group of people. The authors mentioned reference 19. The Epidemiology of Hepatitis C Virus in the Fertile Crescent: Systematic Review and Meta-Analysis, https://doi.org/10.1371/journal.pone.013528. Thuis article shows various analyzes, commenting on HCV prevalence among the general population and  populations at high risk. Their meta-analyzes estimated HCV prevalence among the general population at 0.2% in Iraq (range: 0–7.2%; 95% CI: 0.1–0.3%), 0.3% in Jordan (range: 0– 2.0%; 95% CI: 0.1–0.5%), 0.2% in Lebanon (range: 0–3.4%; 95% CI: 0.1–0.3%), 0.2% in Palestine (range: 0–9.0%; 95% CI) : 0.2–0.3%), and 0.4% in Syria (range: 0.3–0.9%; 95% CI: 0.4–0.5%). Among populations at high risk, HCV prevalence was estimated at 19.5% in Iraq (range: 0–67.3%; 95% CI: 14.9–24.5%), 37.0% in Jordan (range: 21–59.5%; 95% CI: 29.3 –45.0%), 14.5% in Lebanon (range: 0–52.8%; 95% CI: 5.6–26.5%), and 47.4% in Syria (range: 21.0–75.0%; 95% CI: 32.5–62.5%). Genotypes 4 and 1 appear to be the dominant circulating strains. This study shows that Lebanon's border countries have similar HCV prevalence among the general population

  1. 0.2% in Iraq
  2. 0.3% in Jordan
  3. 0.2% in Palestine
  4. 0.4% in Syria

Please, correctly describe the prevalence of HCV prevalence among the general population in Lebanon and  Lebanon's neighboring countries! Everywhere in the text, try to clear up the wrong information and related comments.

Please, include the date of HCV prevalence among populations at high risk in Lebanon and compared it with data from neighboring countries.

Author Response

We thank Reviewer 2 for the valuable comments.

  • Data on HCV prevalence in the general population in Lebanon and the neighboring countries were corrected.
  • Data on HCV prevalence in high-risk groups in Lebanon and the neighboring countries was added.